# A Comparative Analysis of Pretrained Language Models for Text-to-Speech

*Marcel Granero-Moya, Penny Karanasou, Sri Karlapati, Bastian Schnell,*
*Nicole Peinelt, Alexis Moinet, Thomas Drugman*

Alexa AI, Amazon

moymarce@amazon.com

## Abstract

State-of-the-art text-to-speech (TTS) systems have utilized pretrained language models (PLMs) to enhance prosody and create more natural-sounding speech. However, while PLMs have been extensively researched for natural language understanding (NLU), their impact on TTS has been overlooked. In this study, we aim to address this gap by conducting a comparative analysis of different PLMs for two TTS tasks: prosody prediction and pause prediction. Firstly, we trained a prosody prediction model using 15 different PLMs. Our findings revealed a logarithmic relationship between model size and quality, as well as significant performance differences between neutral and expressive prosody. Secondly, we employed PLMs for pause prediction and found that the task was less sensitive to small models. We also identified a strong correlation between our empirical results and the GLUE scores obtained for these language models. To the best of our knowledge, this is the first study of its kind to investigate the impact of different PLMs on TTS.

**Index Terms**: pretrained language models, text-to-speech

## 1. Introduction

The advent of Transformer-based pretrained language models (PLMs) has revolutionized natural language processing (NLP). Initially, Transformers [1] were proposed as an attention-based encoder-decoder architecture for machine translation. But the Transformer decoder was rapidly adopted for text generation with GPT-2 [2] and its successors, and the transformer encoder for natural language understanding (NLU) with BERT [3]. Regarding transformer encoders for NLU, since the emergence of BERT, many BERT-like models have been released with more pretraining data [4, 5], novel pretraining methods [6], more efficient architectures [7], or smaller architectures with faster inference [8, 9, 10, 11]. Generally, these models are trained on a two-step setup. First, they are pretrained from scratch on task-agnostic objectives such as masked language modelling, next sentence prediction, or replaced token detection. Self-supervised pretraining enables the model to learn from large unlabeled corpora like Wikipedia or the Book corpus [12]. Second, they are fine-tuned on a task-specific objective and corpus.

The success of pretrained language models on NLU has inspired research in other fields. In text-to-speech (TTS), several studies have incorporated pretrained language models to Tacotron2 [13], an encoder-decoder TTS system relying on attention. Fang et al. [14] enhance Tacotron2 by adding BERT as a secondary encoder. Thus, the decoder attends to both the subword-level representations generated by BERT and the output of the original Tacotron2 encoder. In a similar fashion, Hayashi et al. [15] compare subword-level and phrase-level representations from BERT-large as extra inputs to Tacotron2

decoder. They conclude that text context helps in generating more natural speech specially at the granularity of subwords. Xiao et al. [16] include Chinese BERT to improve prosody in Tacotron2. They also leverage the pretrained language model to predict pauses between Chinese characters. Alternatively, Xu et al. [17] propose using BERT embeddings to generate context vectors from neighboring sentences to improve prosody modelling. Away from Tacotron 2, Kenter et al. [18] include small versions of BERT in an RNN-based TTS system and highlight the importance of fine-tuning the pretrained model. They present an ablation study on the size of the language model for $F_0$ prediction. More recently, Makarov et al. [19] demonstrate that BERT enhances prosody prediction specially when it is fed with multiple sentences. Karlapati et al. [20] present CopyCat2, a TTS system that learns a prosodic space and subsequently predicts prosody representations based on contextualized subword embeddings from BERT. Similarly, eCat [21] is introduced as a system that predicts prosody with RoBERTa and blocks of normalizing flows.

PLMs have been widely surveyed [22, 23, 24, 25] and compared empirically [26, 27, 28] for various NLP tasks. They are often compared on GLUE [29], an NLU benchmark combining 9 tasks such as natural language inference, sentiment analysis, and sentence similarity. Notwithstanding the numerous works studying the impact of PLMs in NLP and the positive results of PLMs in TTS systems, they have not been surveyed nor compared for TTS tasks, leaving many questions unanswered. First, although several models have outperformed BERT for NLP tasks recently, there is almost no empirical data on how PLMs other than BERT perform in TTS systems. Second, there is little information about the relation between model properties (e.g., size, architecture, and pretraining methodology) and their performance in TTS tasks. Third, there is no comparison of PLMs' performance on NLU and TTS tasks like prosody prediction. This relation would be beneficial to link results of PLMs in NLU to potential findings in TTS.

This work presents a comparative analysis of 15 PLMs for two TTS tasks: prosody prediction and pause prediction. To the best of our knowledge, this is the first study of its kind in the field of TTS. Our contributions are summarized as follows. Firstly, we conduct an experimental analysis of the 15 PLMs for both prosody and pause prediction. Secondly, the results for prosody prediction demonstrate a logarithmic relation between model size and quality, and reveal differences in performance for neutral and expressive prosody. Thirdly, our results on pause prediction indicate that the task is less sensitive to smaller language models. Lastly, we compare our findings with the GLUE scores, highlighting similarities between the performance of PLMs on TTS and NLU.

| Language model | HuggingFace's model name | Num. parameters (M) |
|---|---|---|
| $BERT_{TINY}$ [11] | prajjwal1/bert-tiny | 4.39 |
| $BERT_{MINI}$ [11] | prajjwal1/bert-mini | 11.17 |
| $ELECTRA_{SMALL}$ [6] | google/electra-small-discriminator | 13.48 |
| $MobileBERT$ [10] | google/mobilebert-uncased | 24.58 |
| $BERT_{SMALL}$ [11] | prajjwal1/bert-small | 28.76 |
| $SqueezeBERT$ [9] | squeezebert/squeezebert-uncased | 51.09 |
| $DistilBERT$ [8] | distilbert-base-cased | 65.19 |
| $BERT_{BASE}$ [3] | bert-base-cased | 108.31 |
| $ELECTRA_{BASE}$ [6] | google/electra-base-discriminator | 108.89 |
| $RoBERTA_{BASE}$ [4] | roberta-base | 124.65 |
| $FunnelTransformer_{SMALL}$ [7] | funnel-transformer/small | 130.97 |
| $FunnelTransformer_{INTERMEDIATE}$ [7] | funnel-transformer/intermediate | 177.06 |
| $XLM - RoBERTa_{BASE}$ [5] | xlm-roberta-base | 278.04 |
| $BERT_{LARGE}$ [3] | bert-large-cased | 333.58 |
| $RoBERTA_{LARGE}$ [4] | roberta-large | 355.36 |

Table 1: *Selection of pretrained language models downloaded from HuggingFace [30] and their number of parameters.*

## 2. Methods

This study investigates how different PLMs impact two distinct TTS tasks: prosody prediction and pause prediction. To predict prosody, we use the prosody predictor from eCat [21], an end-to-end TTS model. We delve into the creation and prediction of the latent prosodic space in subsection 2.1. In addition, we analyze pause prediction as a separate task in subsection 2.2. Finally, we present the selected PLMs to be tested on the aforementioned tasks and explain their differences in subsection 2.3.

### 2.1. Prosody prediction

We investigate the impact of PLMs on prosody prediction using eCat [21] because it is a state-of-the-art TTS system that operates in a prosodic latent space. eCat latent space is designed in a way to factor out phoneme and speaker information and capture prosody. It works in a two-step approach where it first learns word-level acoustic and duration-based prosodic representations through an auto-encoder step. In the first step, reference encoders generate the prosodic space based on the mel-spectrogram, and decoders synthesize the speech waveform based on the prosodic representations. The acoustic representations are expected to capture the main prosodic characteristics such as intonation and tone, whereas duration representations are expected to focus on rhythm, stress, and duration. In the second step, a prosody prediction model uses a PLM, followed by Bi-LSTM layers and normalizing flow blocks, to predict these learned prosodic representations from text. The language model produces contextualized word embeddings that capture syntactic and semantic information from text. These embeddings are then fed through Bi-LSTM layers and combined with speaker embeddings to be used as conditions for flow layers. The outputs of the prosody prediction model are used in inference replacing the reference encoder outputs since mel-spectrograms are not available at test time. In this study, we aim to replace the default PLM used in the prosody predictor, i.e. *RoBERTa-base*, with alternative PLMs and evaluate their effectiveness.

### 2.2. Pause prediction

In order to achieve a diverse range of tasks, we examine various PLMs for pause prediction. The objective of pause prediction is to predict the probability of a pause after each word, thereby ensuring that the pauses are accurately placed. Our pausing model consists of a language model (specifically, *RoBERTa-base*), a Bi-LSTM layer, and a dense projection. The model is trained using binary cross-entropy loss between the predicted values and the target binary pausing sequence. Our target binary pausing labels are derived from ground-truth durations, where any silence lasting more than 100ms is considered a pause. This threshold was chosen empirically. Our aim was to annotate anything that is reasonably a silence in terms of audio signal, rather than focusing on pauses from a human perception perspective. During inference, we binarize the predicted probabilities using a threshold previously computed on the development set. This model can be used in eCat to override the duration prosodic representations of pauses. Similar to prosody prediction, we want to compare the pause prediction model with alternative PLMs of different sizes and pretrained with different tasks and data.

### 2.3. Selection of pretrained language models

We select 15 pretrained language models, including the default PLM (*RoBERTa-base*), for both the prosody predictor and the pause prediction model. All language models are BERT-like models, i.e., transformer encoders trained with large corpora of data in a self-supervised fashion. Our selection aims to cover a broad spectrum of model sizes with diverse pretraining or distillation techniques. In Table 1, we provide a list of the pretrained language models, along with their corresponding HuggingFace name and model size. Note that the size refers to the embedding matrix and model architecture weights combined.

We first select medium- and large-size models that target NLU performance. BERT [3] (*BERT-base*, *BERT-large*) was the first pretrained transformer encoder achieving state-of-the-art performance in multiple NLP tasks. It was pretrained on 2 tasks: masked language modelling and next sentence prediction. RoBERTa [4] (*RoBERTa-base*, *RoBERTa-large*) leverages BERT's architecture but is trained longer over more data with longer input sequences and bigger batches. Authors removed the next sentence prediction task, and used masked language modelling dynamically. Overall, RoBERTa outperforms BERT in several NLU benchmarks. XLM-RoBERTa [5] is a multilingual version of RoBERTa. It has the largest embedding matrix out of the selection, consisting of 192 million parameters due to its coverage of 100 languages. Despite this, it is comparable in architecture size to both *BERT-base* and *RoBERTa-base*. ELECTRA [6] (*ELECTRA-base*, *ELECTRA-small*) is trained with replaced token detection instead of masked language mod-

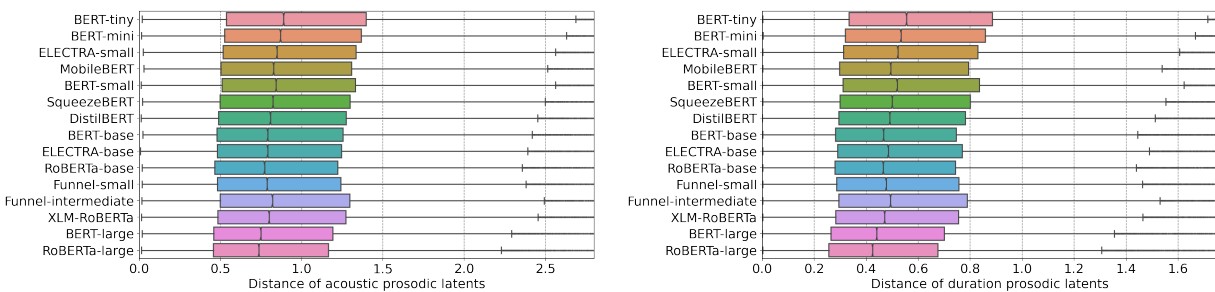

Figure 1: *Euclidean distances between predicted and ground-truth latents for acoustic (left) and duration (right) representations. Language models are shown in ascending order of size.*

elling. A small generator network corrupts the text by replacing tokens and a discriminator is trained to identify the replaced tokens. Authors claim a more efficient training while obtaining better results than previous models. Funnel-Transformer [7] (*Funnel-intermediate*, *Funnel-small*) makes an architectural change by gradually compressing the sequence of hidden states. It is trained with a similar objective as ELECTRA.

Regarding small models, transfer learning from large-scale language models and more efficient architectures have reduced the size and increased speed of language models while retaining performance. DistilBERT [8] is a distilled version of BERT that reduces BERT size by 40%, while preserving 97% of its NLU performance. SqueezeBERT [9] uses grouped convolutions instead of fully-connected layers as they account for a great percentage of FLOPs and latency. MobileBERT [10] is a deep and thin language model that reduces BERT size by 4. It is trained through progressive knowledge transfer from a bigger network. Turc et al. [11] introduce the concept of pretrained distillation to improve the efficiency of knowledge distillation and release several distilled versions of BERT (*BERT-small*, *BERT-mini*, *BERT-tiny*).

## 3. Experimental setup

### 3.1. Data

We conducted experiments on an internal dataset containing recordings of speakers reading excerpts from Wikipedia articles, news articles, conversations, etc. The dataset comprises over 20 hours of speech per speaker, recorded from 4 female speakers of US English. All recordings were sampled at 24kHz and documents are not shared across data splits (train/dev/test). All data splits contain utterances with diverse levels of expressivity. We divided them into utterances with more neutral speech such as discourse and utterances with highly expressive speech such as quotes. Differences in expressivity were taken into account during evaluations (see Section 4.1).

### 3.2. System configuration

In this work, we did not tune hyperparameters for every model that was trained. We used the same learning rate and training steps across all language models to maintain simplicity and consistency in our experiments. However, we fine-tuned language models of diverse sizes and pretraining techniques. This may have resulted in an advantage for the *RoBERTa-base* model compared to the other models, as the hyperparameters were tuned specifically for this pre-trained language model. For instance, large language models may have been under-trained.

For the prosody predictor, we used the training hyperparameters from the original eCat [21], which were tuned for *RoBERTa-base*. The Bi-LSTM layer had a hidden dimension of 512. To train the pause predictor, we used a batch size of 32, Adam optimizer with a learning rate of $1e$-5 in the language model and $1e$-4 in the rest of the model, with betas of 0.9 and 0.98. The model was trained for 50k steps.

We investigated the effect of different language models on both TTS tasks. First, we trained the prosody predictor with the 15 PLMs listed in Table 1. Second, we trained and compared the pause prediction model with a subset of 12 PLMs. *BERT-small*, *Funnel-small*, and *Funnel-intermediate* were excluded due to implementation issues. In both setups, the prosody predictor and pause prediction model were trained end-to-end, meaning that the losses were back-propagated into the language models to fine-tune textual representations.

Prosody prediction and pause prediction tasks required the use of word-level tokens. However, most language model tokenizers operate at the word-piece level, breaking words into smaller subword units. To bridge this gap, we downsampled the contextualized word-piece embeddings produced by the language model. Specifically, we averaged the word-piece embeddings that aligned with the same word-level tokens, thus obtaining word-level embeddings.

## 4. Results

### 4.1. Language models for prosody prediction

The prosody predictor was trained with every language model from Table 1. To measure the accuracy of our prosody prediction model objectively, we calculated the Euclidean distance between the predicted latent samples and the ground-truth ones. We generated the ground-truth prosody representations by passing the ground-truth mel-spectrogram through a reference encoder. Generally, the closer the predicted latent sample is to the ground-truth latent, the more coherent and appropriate it is. It is worth noting that during training, we did not use Euclidean distance, as we employed negative log-likelihood on normalizing flows instead. Figure 1 shows Euclidean distance, for acoustic (left) and duration (right) latents. On the y axis, language models are sorted by descending size. Results manifest a negative correlation between size and distance; prosody predictors with bigger language models reduce the distance to ground-truth representations for both acoustic and duration latents. In other words, bigger language models seem to better predict prosody latents.

In addition, we conducted subjective evaluations on syn-

| Alternative LM | eCat with alt. LM | eCat with RoBERTa-base | Recordings | Gap reduction (%) | p-value | Significance |
|---|---|---|---|---|---|---|
| $BERT_{TINY}$ | $73.86 \pm 16.37$ | $74.99 \pm 15.37$ | $78.32 \pm 14.58$ | -33.73 | 0 | TRUE |
| $BERT_{MINI}$ | $72.77 \pm 16.35$ | $73.64 \pm 15.79$ | $77.31 \pm 15.31$ | -23.78 | 0.0026 | TRUE |
| $MobileBERT$ | $73.25 \pm 18.32$ | $74.15 \pm 18.14$ | $78.01 \pm 17.40$ | -23.19 | 0.0019 | TRUE |
| $DistilBERT$ | $69.04 \pm 19.75$ | $69.83 \pm 19.36$ | $75.96 \pm 17.87$ | -12.86 | 0.0017 | TRUE |
| $RoBERTa_{LARGE}$ | $72.23 \pm 16.36$ | $72.67 \pm 15.76$ | $75.98 \pm 15.30$ | -13.14 | 0.0914 | FALSE |

Table 2: *MUSHRA mean scores. Every row is a MUSHRA evaluation with 3 systems: eCat with the alternative LM from the first column, eCat with RoBERTa-base, and recordings. Gap reduction is the difference between RoBERTa-base and the alternative language model relative to the difference between recordings and RoBERTa-base.*

thetic speech. We launched 5 MUSHRA evaluations comparing recordings, speech from vanilla eCat (i.e., with *RoBERTa-base* in the prosody predictor), and speech generated by eCat with an alternative language model in the prosody predictor. The alternative language models were selected keeping a broad range of sizes, namely, *RoBERTa-large*, *DistilBERT*, *Mobile-BERT*, *BERT-mini*, and *BERT-tiny*. All evaluations were conducted with 24 testers and the same 80 utterances, consisting of 20 utterances per speaker. Approximately half of the samples per speaker comprised neutral speech while the other half consisted of highly expressive speech. The MUSHRA question was about the naturalness of the voice. We expect any changes to be due to prosody since we are only modifying the prosody predictor, i.e., only changing the predicted latents. Table 2 summarises results for each MUSHRA evaluation. It displays mean MUSHRA scores for 3 systems: recordings, vanilla eCat (with *RoBERTa-base*), and eCat with an alternative language model in the prosody predictor. We show the reduction of gap between vanilla eCat and recordings; negative values indicate a gap increase. Moreover, a two-sided t-test was computed to compare vanilla eCat against eCat with an alternative language model. The last two columns show the $p$-$value$ and whether it was statistically significant given a significance level of $\alpha = 0.05$.

As a result, all alternative models perform statistically significantly worse than *RoBERTa-base* except for *RoBERTa-large*. Similar to objective metrics, there is a relation between size and gap reduction. Smaller models have worse degradation than larger models. Notably, the fourth MUSHRA evaluation yielded the lowest scores for eCat with *DistilBERT* and eCat with *RoBERTa-base*. We attribute this outcome to the fact that this specific evaluation took place several weeks after the rest of the evaluations and involved a different set of listeners.

Furthermore, we examined MUSHRA results for neutral and expressive speech separately. Our systems were rated worse on expressive speech and received scores close to recordings for neutral speech. There were no statistically significant differences between *RoBERTa-base* and alternative models on neutral speech, except for *BERT-tiny*, the smallest one. On the other hand, *RoBERTa-base* was statistical significantly better than all alternative systems on expressive speech. We hypothesize that prosody prediction on expressive speech is more challenging than in neutral speech because of its variance in pitch, tone, and speed. Prosody predictors with small language models were able to capture the expressivity involved for neutral speech but failed to reproduce expressive prosody. Therefore, we find more suitable to use large language models for expressive speech, whereas for neutral speech medium-size models would suffice.

Given the objective metrics and subjective evaluations presented, we identify a relation between size and prosodic quality. This relation is manifested on objective metrics as shown in Figures 2a and 2b. We consider *Funnel-intermediate* an out-

lier as it has architectural changes and might have been undertrained. Subjective metrics demonstrate a similar trend in Figure 2c. Overall, the findings suggest that the quality of prosody improves in a logarithmic fashion as the size of language models increases. While our results for prosody prediction rely on eCat, we believe that the conclusions presented in this section will be applicable to other architectures utilizing prosodic latent spaces.

### 4.2. Language models for pause prediction

The pause prediction model was trained with 12 language models including *RoBERTa-base*. We trained the language models in the pause prediction model to predict pause probabilities after each word. Once trained, a threshold was obtained on the dev set. We measured precision, recall, and F-scores for the binarized pause predictions on the test set. Table 3 presents results for every pause prediction model trained. It shows precision, recall and $F$-0.5 and $F$-1 scores. $F$-0.5 is more aligned with our objectives as we believe that failing to predict a pause is not as bad as predicting a wrong pause. Results indicate a relation between size and performance as larger models like *RoBERTa-large* perform better than smaller ones. Actually, the smallest models (i.e., *BERT-tiny* and *BERT-mini*) were substantially behind the rest, but from *ELECTRA-small* until *RoBERTa-large* there was a slighter performance increment. Thus, we hypothesize that there is a minimum size required, e.g., *ELECTRA-small*'s size, to perform on par with medium-size models like *RoBERTa-base*. We also attribute the outstanding results of both *ELECTRA* versions to their discriminative pretraining task, replaced token detection, which makes the model learn more efficiently and robustly from all input tokens at the same time.

In conclusion, we believe that predicting pauses in English only requires a high-level understanding of language, as the task ambiguity is limited, and small models like *ELECTRA-small* are already able to capture it. Compared to prosody prediction, we surmise that there is an order of complexity between the tasks. Predicting prosody for expressive speech would be the most challenging task, followed by predicting prosody for neutral speech. Pause prediction would be less challenging than predicting prosody. Therefore, small language models could replace larger ones as they perform similarly well. It's important to note that our findings are specific to the English language, and further research is required to determine whether the conclusions hold true for other languages.

## 5. Further analysis

### 5.1. Relation with GLUE scores

We investigated whether there are comparable trends in language models performance for TTS and NLU tasks. We compared experimental results for TTS obtained in this study with

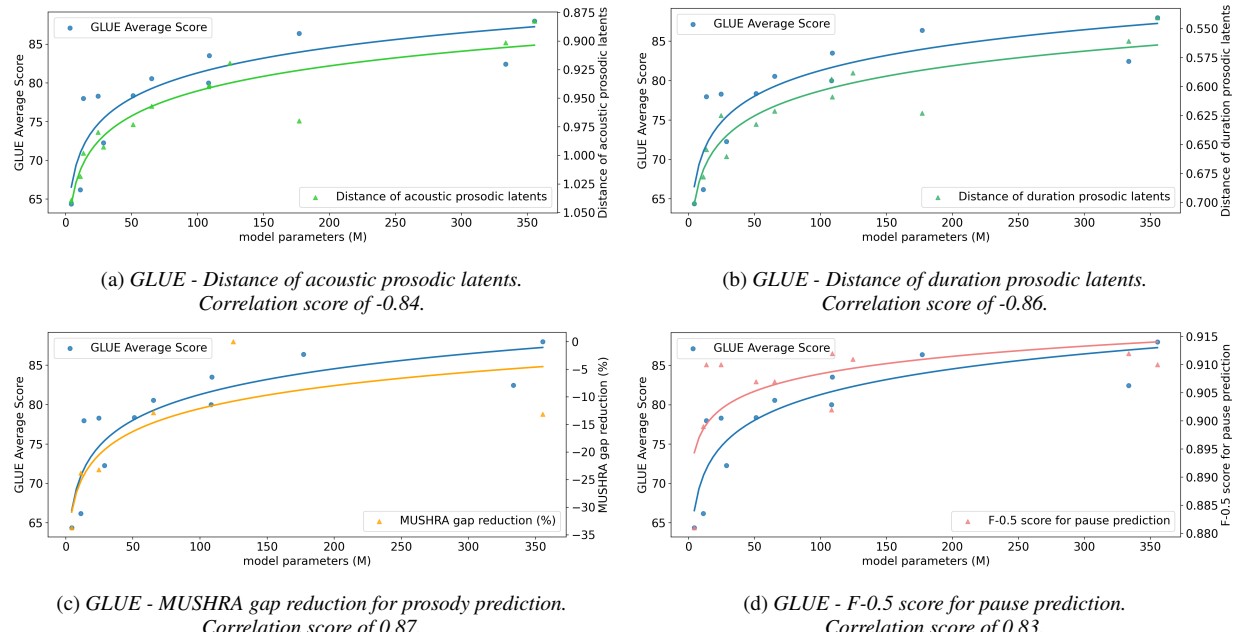

(a) *GLUE - Distance of acoustic prosodic latents. Correlation score of -0.84.*

(b) *GLUE - Distance of duration prosodic latents. Correlation score of -0.86.*

(c) *GLUE - MUSHRA gap reduction for prosody prediction. Correlation score of 0.87.*

(d) *GLUE - F-0.5 score for pause prediction. Correlation score of 0.83.*

Figure 2: *Comparison of GLUE scores with distance of (a) acoustic and (b) duration prosodic latents, (c) MUSHRA gap reduction, and (d) F-0.5 score for pause prediction. Logarithmic regressions are included for every metric.*

| Language Model | Precision | Recall | F-0.5 | F-1 |
|---|---|---|---|---|
| $BERT_{TINY}$ | 0.895 | 0.828 | 0.881 | 0.860 |
| $BERT_{MINI}$ | 0.920 | 0.825 | 0.899 | 0.870 |
| $ELECTRA_{SMALL}$ | 0.934 | 0.828 | 0.910 | 0.877 |
| $MobileBERT$ | 0.931 | 0.832 | 0.910 | 0.879 |
| $SqueezeBERT$ | 0.938 | 0.802 | 0.907 | 0.865 |
| $DistilBERT$ | 0.929 | 0.831 | 0.907 | 0.877 |
| $BERT_{BASE}$ | 0.916 | 0.852 | 0.902 | 0.883 |
| $ELECTRA_{BASE}$ | **0.941** | 0.811 | **0.912** | 0.871 |
| $RoBERTa_{BASE}$ | 0.933 | 0.831 | 0.911 | 0.879 |
| $XLM\text{-}RoBERTa$ | 0.925 | 0.827 | 0.904 | 0.873 |
| $BERT_{LARGE}$ | 0.933 | 0.837 | **0.912** | 0.882 |
| $RoBERTa_{LARGE}$ | 0.924 | **0.858** | 0.910 | **0.889** |

Table 3: *Results for the pause prediction model with different language models. Models are shown in ascending order of size.*

scores on GLUE test set. GLUE [29] is a benchmark composed of 9 diverse language understanding tasks. Note that we collected GLUE scores from original publications of language models, while the language models used in this work were downloaded from HuggingFace. We compared the GLUE scores with the metrics reported for prosody prediction and pause prediction, and computed correlation scores. Figure 2 shows the comparison of GLUE scores with (a) distance of acoustic prosody latents to ground-truth, (b) distance of duration prosody latents to ground-truth, (c) MUSHRA gap reduction, and (d) F-0.5 score for pause prediction. We computed the correlation coefficient between each pair of score sequences, and plotted a logarithmic regression. The comparison reveals a robust correlation between GLUE scores and the reported metrics, with all absolute correlation scores exceeding 0.8. These results suggest that language models enhance prosody and pause prediction similarly to natural language understanding tasks, as demonstrated by the comparison

with GLUE scores. Additionally, the metrics exhibit a logarithmic relationship with size, meaning that an increase in size generally corresponds to an increase in quality. Based on the quality-size relationship and the correlation with GLUE scores described earlier, it seems reasonable to estimate the performance of BERT-like language models for prosody and pause prediction by considering architectural size and their GLUE score.

### 5.2. Model throughput analysis

Batch inference time was measured for all prosody predictors trained with different language models. We used an NVIDIA Tesla V100 SXM2 32GB GPU with CUDA 11.6 and Pytorch 1.11.0+cu113. 8 iterations were run in a test set of 2500 samples with a batch size of 32 samples. For a better estimate, the first batch was discarded to avoid GPU initialization overhead, and the last one was also discarded as its batch size was smaller.

Figure 3 displays batch inference time for each language model. All values are relative to the mean batch inference time of the prosody predictor with *RoBERTa-base*. The figure on the right depicts mean batch inference time compared to the prosody predictor size. The results demonstrate a linear relation between batch inference time and models size besides a few outliers. For instance, *MobileBERT* was slower than models with similar size. In fact, it was slower than what was reported on the original paper. We attribute this gap to the fact that the model was not the original, but instead was uploaded to HuggingFace by an external user. On the other hand, even though *XLM-RoBERTa* doubled in total size, its inference time remained similar to *BERT-base*. This is because their architecture sizes are identical, and the increase in size is attributed to the embedding matrix as *XLM-RoBERTa* comprises a wide multilingual vocabulary.

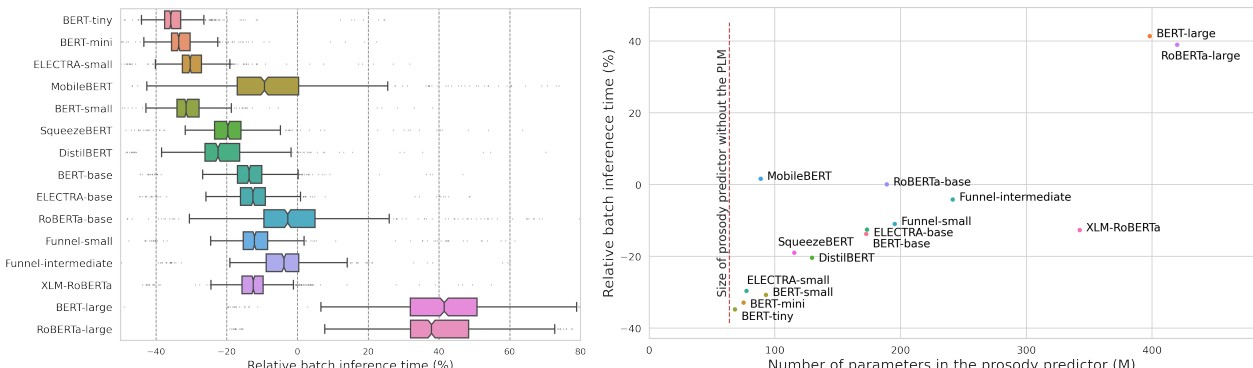

Figure 3: *Batch inference time for the prosody predictor, which varies depending on the language model used. All the times are presented relative to the mean inference time for the prosody predictor when using RoBERTa-base.*

## 6. Conclusion

This work presents a comparative analysis between 15 PLMs for two TTS tasks in US-English: prosody prediction and pause prediction. We evaluated language models performance on prosody prediction with objective and subjective metrics. Results indicate a logarithmic relation between size and quality. We expanded to pause prediction, where small language models performed on par with larger ones. Moreover, we compared this study's results with GLUE scores and identified similar trends. Thus, improvements in language models for NLU can inform future developments and improvements in TTS. To the best of our knowledge, this is the first study of this kind for TTS.

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
