# OpenReview forum: "A Comparative Analysis of Pretrained Language Models for Text-to-Speech"
_Interspeech.org/2023/Workshop/SSW — SSW12_

### Official Review · Reviewer_z5ur · 2023-06-02
**This paper investigates the performance of using various pre-trained language models (PLMs) on the quality of synthetic speech.**

**Rating:** 7
**Confidence:** 4

**Review:**

This paper investigates the performance of using various pre-trained language models (PLMs) on the quality of synthetic speech. The paper assumes an underlying TTS architecture that exploits a learned latent space (learned from ground truth acoustic data ) and then uses PLM features to predict such latent features at run-time. Since we have seen PLMs being deployed in the TTS literature to harness gains in various ways across various architectures, this paper provides good value in its systematic analysis of a substantial number of such PLMs while keeping the architecture constant, and focusing instead on different families, and model sizes.

The results about the effect of model size on the quality of TTS seem warranted by the experiments. My main objection has to do with the more specific claim that these gains come from an improvement in prosody, which the PLM are helping to better model. This simply may be due to the fact that the underlying eCat architecture is described in insufficient detail in Sec 2.1 (as it seems to be the subject of a separate submission to Interspeech under review). For instance:

* “In the first step, reference encoders generate the prosodic space based on the mel-spectrogram, and decoders synthesize the speech waveform based on the prosodic representations. The acoustic representations are expected to capture intonation and tone, whereas duration representations focus on rhythm, stress, and duration.” – This tells us there is a mel-to-latent-representation component followed by a latent-to-waveform component(s). As described, there is nothing that suggests that this latent representation is prosodic (unless it is trained with an objective that enforces this).  Fig 1 (a) and (b) sorts out “prosodic latents” from “duration latents”, so perhaps there is some interpretability to this latent space that isn’t clear from the paper.

* “The acoustic representations are expected to capture intonation and tone” – Has this been verified in some way?

* “ it first learns word-level acoustic and duration-based prosodic representations through an auto-encoder step…” -- Prosody involves more fine-grained scales than word level. Is the idea here that the eCat architecture (in particular the decoder) is able to reconstruct this finer structure from coarser descriptors, and that using a PLM provides better guidance?

Furthermore, the evaluation methodology (MUSHRA) doesn’t seem to target prosody specifically (unless crucial details about what the listeners were asked have been omitted). So the following conclusion about prosodic improvements also seems unwarranted: “Subjective metrics demonstrate a similar trend in Figure 2c. Overall, the findings suggest that the quality of prosody improves in a logarithmic fashion as the size of language models increases”

These issues don’t change the main finding about the effect these PLMs have on (a) the internal latent representation (Fig 1) or (b) how they help with perceptual quality in a general way (Table 2). But the interpretation of *how* these PLMs are helping seems an extrapolation from the evidence (as presented). It would be good to see the authors provide the necessary evidence to support these claims, or else rephrase the findings in a more neutral way, as I think the findings are still valuable.




Some other comments / clarifications:
- How did you reconcile the different (sub-word) tokenizations of the various PLMs with the word-level processing of the eCat encoder?
- Why were 2 PLMs omitted from the pausing experiments? And which?
- Table 2: Can you explain the considerable drop in quality of the eCat w/RoBERTa-base when paired with the DistilBERT alternative (69.83 vs remaining values witin that column)? This drop is far more significant than, say, the BERT-tiny vs RoBERTa difference. If the samples are constant, we should expect the within-model performance to be more stable. Were different testers allocated to different comparisons so that this can be attributed to different listeners?
- There are a few capitalization issues in the bibliography: tts-> TTS / bert -> BERT / etc.

---

### Official Review · Reviewer_dH8A · 2023-06-05
**Review of A Comparative Analysis of Pretrained Language Models for Text-to-Speech**

**Rating:** 8
**Confidence:** 5

**Review:**

This manuscript embarks on a comprehensive analysis of the influence of pretrained language models (PLMs) on the quality of neural Text-to-Speech (TTS) systems, especially concerning prosody and pause predictions. The study is distinguished by its ambitious scope, evaluating an array of 15 BERT-like models that vary in size and complexity.

The experiment's methodology involves training TTS voices on four different 20-hour corpora, each featuring American English-speaking female voices. The speech data was divided into neutral and expressive categories, an approach that offers a nuanced understanding of the TTS systems' performance under varying emotional nuances.

In pursuit of objective evaluation, the authors used the prosodic latent space created by eCat and considered any silence exceeding 100 ms as a pause for pause predictions. These metrics were then used to juxtapose the TTS samples with the corresponding ground truth recordings, providing a robust comparison.

A MUSHRA listening test was utilized for subjective evaluation, comparing baseline eCat TTS samples, those generated with the support of PLMs, and the original recordings. Interestingly, the findings suggest that larger PLMs outperform their smaller counterparts in the neural speech category, being closer to the ground truth recordings. However, all models fell short in replicating expressive speech, with smaller models struggling more.

This study also revealed a strong correlation between the TTS metrics and the General Language Understanding Evaluation (GLUE), further reinforcing the findings.

Strengths:
This paper delivers a highly ambitious and critical evaluation, focusing on improving prosody in TTS—an area that undoubtedly deserves attention.

Limitations:
The paper contains some ambiguities in terms of the prosodic comparisons and the selection criteria for the pause task. Further clarification in these areas could enhance the comprehensibility of the research and its eventual applicability.

Some questions to the authors
Your definition of "expressive" is somewhat ambiguous. You mention “utterances with highly expressive speech such as quotes”, but it's unclear what exactly qualifies an utterance as 'expressive'. Could you provide more context? Does it relate to the inclusion of emotive words such as "I love/hate something", or is it related to the way it is written, for instance, “that was strange, he said in a surprised voice"? How does the Text-to-Speech system distinguish between neutral and expressive utterances?

Regarding pause detection, your methodology tags any silence over 100 ms as a pause. Could you elaborate on the reason behind selecting this particular threshold? Many studies use a threshold of 200 ms, and some suggest that the shortest perceptible pause is around 140 ms. Have you considered utilizing a ToBI-like system, where pauses of different lengths are clustered into break indices? There's a significant perceptual and production difference between a 100 ms pause and an 800 ms pause.

In the section discussing the objective comparison of prosodic realization, you mention calculating the Euclidean distance between the predicted latent samples and the ground-truth ones. Am I correct in assuming this distance was calculated between the held-out data from the original recordings and the synthesized text? Were the same text passages used across all four TTS corpora? If so, what would the Euclidean distance in the prosodic latent space between pairs of human recordings be? Could this value provide any insight into the challenges associated with comparing prosodic realizations?

---

### Decision · Program_Chairs · 2023-06-14

**Decision:**

Accept

**Comment:**

SSW2003 received 45 papers. The acceptance rate is 82%. We are pleased to inform you that your paper has been accepted by the SSW2023 Program Committee. Please read the reviews carefully and submit your camera-ready paper by June 28th. Most reviewers performed a detailed review. Please answer to their questions and consider their comments. Note that camera-ready papers are credited with one extra page to allow authors to consider reviewers’ suggestions. So max 7 pages in total including figures & refs.
The deadline for submitting the revised version (with full non-anonymized authors and refs!) is 28th June.